# Recent Progress in the Rational Design of Biothiol-Responsive Fluorescent Probes

**DOI:** 10.3390/molecules28104252

**Published:** 2023-05-22

**Authors:** Wenzhi Xie, Jinyu Jiang, Dunji Shu, Yanjun Zhang, Sheng Yang, Kai Zhang

**Affiliations:** 1Hunan Provincial Key Laboratory of Cytochemistry, School of Chemistry and Chemical Engineering, Changsha University of Science and Technology, Changsha 410114, China; 2Department of Chemistry, School of Basic Medical Sciences, Southwest Medical University, Luzhou 646000, China; 3Laboratory of Chemical Biology &Traditional Chinese Medicine Research, Ministry of Education, College of Chemistry and Chemical Engineering, Hunan Normal University, Changsha 410081, China; djshu@hunnu.edu.cn

**Keywords:** biothiols, fluorescent probe, reaction mechanism, discriminated detection, reversible detection, specific imaging

## Abstract

Biothiols such as cysteine, homocysteine, and glutathione play significant roles in important biological activities, and their abnormal concentrations have been found to be closely associated with certain diseases, making their detection a critical task. To this end, fluorescent probes have become increasingly popular due to their numerous advantages, including easy handling, desirable spatiotemporal resolution, high sensitivity, fast response, and favorable biocompatibility. As a result, intensive research has been conducted to create fluorescent probes for the detection and imaging of biothiols. This brief review summarizes recent advances in the field of biothiol-responsive fluorescent probes, with an emphasis on rational probe design, including the reaction mechanism, discriminating detection, reversible detection, and specific detection. Furthermore, the challenges and prospects of fluorescence probes for biothiols are also outlined.

## 1. Introduction

Biological thiols, known as biothiols, are essential in a variety of important physiological and pathological processes in living cells and organisms [1,2]. Glutathione (GSH), cysteine (Cys), and homocysteine (Hcy) are the most relevant biothiols, and they have similar structures that can transform into each other within cells. GSH is the most abundant non-protein thiol in cells, ranging from 1 to 10 mM, and is closely related to cellular redox homeostasis, heterogeneous metabolism, signal transduction, and gene regulation. Abnormal expression of GSH is associated with various disorders, including Parkinson’s disease, Alzheimer’s disease, liver injury, and cancer [3,4,5]. Cys, with a concentration ranging from 30 μM to 200 μM in cells, is a precursor of GSH and a strong binding site for transition metals such as copper, lead, cadmium, and mercury. Cys plays important roles in protein synthesis, neuroprotection, and redox homeostasis. Overexpression of Cys would lead to cardiovascular and nervous system diseases, whereas Cys deficiency might result in liver and kidney injury, growth retardation, and other diseases [6,7,8]. Hcy has the lowest general concentration among these three biothiols (5–12 μM), and abnormal levels of Hcy could lead to neurological and cardiovascular diseases [9,10,11]. Therefore, developing reliable and effective methods for real-time monitoring of cellular biothiols is of great significance in order to provide crucial information for the investigation of their functions, thereby improving early diagnosis and therapeutic interventions for related diseases [12,13].

Capillary electrophoresis, electrochemical assays, high-performance liquid chromatography (HPLC), and mass spectrometry have all been described as detection methods for the determination of biothiols [14,15,16,17]. However, these traditional methods frequently encounter expensive costs, cumbersome preconditioning procedures, and massive samples, and they are not competent for real-time detection of biothiols. Fluorescent probes have attracted considerable attention in recent decades due to their unique advantages, including easy operation, desirable spatiotemporal resolution, high sensitivity, fast response, and favorable biocompatibility [18,19,20,21,22]. Therefore, over the last 20 years, a large number of biothiol fluorescent probes have been developed, serving as valuable tools for understanding the physiological and pathological functions of biothiols in biological systems [23].

Recently, several excellent reviews on fluorescent sensing of biothiols have been published, covering various perspectives such as fluorescence materials, emission signals, and specific detection [24,25,26,27,28]. Nonetheless, there is a lack of summaries focusing on the design strategies of biothiol-activated small molecular fluorescent probes. Furthermore, new types of probes that enable discriminated, reversible, subcellular detection of biothiols have emerged in the last four years. Thus, a description of the latest research progress is meaningful for junior researchers to comprehend the sensing and design principles of these probes. In this brief review, we will present recent advances in the rational design of biothiol fluorescent probes, categorized according to reaction mechanism, specific imaging, reversible response, and discriminating detection (Figure 1). Due to the page limit, only a few works will be discussed in depth to provide valuable guidance for the creation of biothiol fluorescent probes. Additionally, we will outline the current challenges and prospects to inspire innovation in biothiol fluorescent probes for bio-applications.

## 2. Reaction Mechanisms of Biothiol-Responsive Fluorescent Probes

Fluorescence probes are typically designed by modifying fluorophores with reactive motifs that allow them to fluorescently respond to analytes of interest through several signal transduction mechanisms. These mechanisms include photoinduced electron transfer (PET), twisted intramolecular charge transfer (TICT), through bond energy transfer (TBET), Förster resonance energy transfer (FRET), intramolecular charge transfer (ICT), excited-state intramolecular proton transfer (ESIPT), aggregation-induced emission (AIE), and others [29,30,31,32,33,34,35]. Thus far, different reaction types have been involved in the rational design of fluorescent probes for biothiol sensing, taking advantage of the strong nucleophilicity and reducibility of biothiols. We began this review by introducing the typical reaction mechanisms of recently reported biothiol-responsive fluorescent probes, which utilize various thiol-triggered reactions such as thiolysis reactions, Michael addition reactions, reduction reactions, and aromatic nucleophilic substitution reactions (SNAr).

### 2.1. Biothiol Fluorescent Probes Based on Thiolysis Reactions

Dinitrobenzene sulfonyl (DNBS) derivatives are highly electron-deficient due to the presence of two electron-withdrawing nitro groups, allowing them to efficiently quench the fluorescence by PET processes. In the presence of nucleophilic biothiols, DNBS derivatives can be cleaved by thiolysis reactions, leading to restored fluorescence (Figure 1) [36]. Therefore, a host of biothiol fluorescent probes have been designed by equipping DNBS with different fluorophores. To enable biothiol fluorescent probes suitable for bio-applications, several characteristic properties of fluorophores are required, such as a large Stokes shift, long-wavelength emission/excitation, intense absorption and emission, excellent solubility, low cytotoxicity, high stability, and so on.

For example, taking advantage of 2,7-Naphthyridine derivatives with large Stokes shifts, high quantum yields, and good photo-stabilities, the She group used a 2,7-Naphthyridine derivative as a fluorophore to develop a thiolysis-based fluorescent probe **1** (**AND-DNBS**) for biothiol detection, with the dinitrobenzene sulfonyl (DNBS) moiety serving as the sensing site [37]. As depicted in Figure 2, probe **1** responded to biothiols in a turn-on manner: DNBS was broken off from probe **1** by a thiolysis reaction to remove the PET effect, resulting in the formation of **AND-OH**, with favorable performances including fast response time (45 s), desirable fluorescence enhancement (100 fold), and excellent Stokes shift (227 nm). Probe **1** showed a high quantum yield (0.36). With favorable membrane permeability and low cytotoxicity, probe **1** was successfully used to visualize endogenous and exogenous GSH in living A549 cells and zebrafish.

Traditional organic dyes usually encounter aggregation-caused quenching (ACQ) in biological systems due to their poor water solubility, which hinders their applicability in fluorescence imaging. To address this issue, Tang proposed the concept of aggregation-induced emission (AIE) for the first time [38]. AIE dyes exhibit intensive fluorescence after aggregating in high-concentration aqueous solutions but display negligible emission in low-concentration solutions. Furthermore, aggregation in a biological system can even avoid the dispersion of free dyes, improve the photostability of dyes, and promote the signal-to-noise ratio and sensitivity of detection. Thus far, AIE-based fluorescent probes have drawn considerable research interest in the last 20 years [39,40]. The Wang group reported a thiol-responsive fluorescence probe **2** (**DNBS-HCA**) using an AIEgen hydroxychalcone (HCA) as the reporting unit [41]. The hydroxyl group of HCA was caged with the DNBS moiety to block the ESIPT process, thus quenching the fluorescence. After the thiolysis reaction of probe **2** by biothiols, the recovery **HCA** exhibited remarkable AIE and ESIPT characteristics with long-wavelength red fluorescence emission at 600 nm and a favorable Stokes shift of 140 nm (Figure 2). Probe **2** was then applied to develop indicator papers for biothiol detection, and the study demonstrated that the indicator papers could afford desirable feasibility and sensitivity for the quantitative detection of biothiols (0.1–1 nM). Finally, probe **2** was used to visualize the variation of cellular biothiols, and the red emission got weaker after NEM treatment, indicating the ability to detect biothiols in vivo.

Yuan et al. reported another thiolysis-based probe **3** (**M-OH-SO_3_**) with a dicyanoisophorone-derived fluorophore [42]. Probe **3** exhibited almost no fluorescence due to the intramolecular twisted internal charge transfer (TICT) and shielding effect of the DNBS group [30]. After the reaction with sulfhydryl groups, probe **3** could release the quencher to generate the dicyanoisophorone dye **M-OH-OH**, resulting in elevated emission in the yellow channel (575 nm) under 400 nm excitation in the presence of GSH. Intriguingly, the thiolysis product **M-OH-OH** can further embed into the hydrophobic cavity of human serum albumin (HSA) to produce an augment of redshifted fluorescence in the red channel (660 nm) under excitation at 500 nm due to the blocked TICT (Figure 3A,B). Because of its dual-channel spectral response capability, probe **3** was utilized to simultaneously monitor variations of drug-induced HSA and endogenous/exogenous GSH in HepG2 cells by fluorescent imaging.

### 2.2. Biothiol Fluorescent Probes Based on Michael Addition Reactions

It is well known that thiols can undergo Michael addition reactions with various unsaturated groups such as aldehydes, ketones, maleimides, squaraines, and malonitriles. A variety of outstanding Michael acceptors were involved in the construction of fluorogenic probes for biothiol sensing. The Chen group, for instance, reported a rhodol-derived probe **4** (**FRB-T**, Figure 3) for detecting biothiols in living cells and labeling sulfhydryl-containing proteins [43]. A 2-cyclopentenone group was decorated onto rhodol-derived as a responsive site, allowing the 1,4-Michael addition between the *α*,*β-*unsaturated ketone and free sulfhydryl, and the subsequent ring-opening reaction led to the formation of rhodol in ketone form, resulting in an enhanced fluorescence at 557 nm. In vitro tests revealed that probe **4** was extremely sensitive and selective for biothiols. Furthermore, biothiols in HeLa cells were visualized by probe **4**, and fluorescence was enhanced after co-incubation with Cys or GSH, whereas attenuated fluorescence was observed with *N*-Ethylmaleimide (NEM) as a scavenger for biothiols. Finally, probe **4** was also utilized to label sulfhydryl proteins, giving different fluorescence signals between sulfhydryl proteins and non-sulfhydryl proteins. These results indicated that probe **4** could be a robust tool to track biothiols in vivo and label sulfhydryl-containing proteins.

By connecting the chromene motif and methylene blue with a carbamate linker, the Yin group created a Michael addition-based fluorescent probe **5** (Figure 4) [44]. Figure 4 shows that methylene blue could be released in the presence of biothiols via a cascade reaction including 1,4-Michael addition on the *α*,*β*-unsaturated ketone, ring opening of pyran, and self-immolation of carbamate, resulting in a considerable increase in the fluorescence at 698 nm under illumination at 600 nm. Owing to its long-wavelength absorption and emission, probe **5** was expected to possess advantages such as deep tissue penetration, minimal phototoxicity, and a high signal-to-noise ratio (S/N). Probe **5** demonstrated a fast response (~150 s), long-wavelength emission, and high chemical stability under physiological circumstances. Probe **5** with minimal cytotoxicity was also successful in identifying biothiols in living cells and mouse models, with considerably higher levels of biothiols observed in drug-resistant cancer cells compared to the corresponding cancer cells, indicating that biothiols might facilitate chemotherapy resistance in cancer cells.

### 2.3. Biothiol Fluorescent Probes Based on Reduction Reactions

The disulfide bond (-S-S-) could be reduced by thiols into two sulfhydryl groups. Thus, disulfide cleavage reactions were also utilized in the design of biothiol-responsive fluorescent probes. Wang et al. designed and synthesized a novel disulfide reduction-based ratiometric fluorescent probe **6** (**RB-GSH**) for biothiol imaging (Figure 5) [45]. Probe **6** was constructed on a Förster resonance energy transfer (FRET) scaffold using oxanthrene as the acceptor and BODIPY as the donor, with the disulfide linker between the energy dyad serving as the reaction site. Upon excitation with the absorption of BODIPY at 480 nm, a near-infrared (NIR) fluorescent emission from oxanthrene at 656 nm was identified in the fluorescence spectra of probe **6** due to the resonance energy transfer between donor and acceptor, whereas the fluorescence emission from the BODIPY donor at 511 nm was relatively faint. In the presence of reductive biothiols, the disulfide bond was cleaved by a reduction reaction to effectively inhibit the FRET process, thus leading to a gradual increase at 511 nm and a slow decrease at 656 nm. The fluorescence ratio (I_512nm_/I_656nm_) displayed an excellent linear relationship with GSH concentration from 10 μM to 100 μM. Therefore, this fluorescent probe **6** could determine the biothiols based on the fluorescent signals from two different wavelengths. In contrast to turn-on fluorescent probes, which rely on changes in fluorescence intensity to determine biothiols, ratiometric fluorescent probes employ two signals that act as internal references to each other during the detection process. This approach can overcome potential disturbances caused by factors such as microenvironment (pH, polarity, viscosity, and so on), distribution and photobleaching of probes, and intensity of light sources, thereby enabling probe **6** to accurately detect intracellular GSH in a ratiometric manner.

### 2.4. Biothiol Fluorescent Probes Based on SNAr Reactions

In addition to thiolysis reactions, Michael addition reactions, and disulfide reduction reactions, aromatic nucleophilic substitution reactions (SNAr) were another powerful tool to develop biothiol probes based on their nucleophilicity [46]. Generally, the SNAr-based probes undergo two successive reactions in the response processes, including an intermolecular nucleophilic substitution and an intramolecular nucleophilic substitution, enabling the specific detection of GSH over Hcy/Cys, which we will discuss in the sections below.

## 3. Design Strategies for Specific Fluorescent Probes for Intracellular Biothiols

Traditional thiol-responsive fluorescent probes have difficulty distinguishing different biothiols from each other due to their structural similarities. So far, smart fluorescent probes capable of selectively detecting and imaging GSH or Cys/Hcy have attracted intensive research interest to obtain more detailed information for studies on physiological processes and disorders related to one specific biothiol. Despite the fact that it is a significant challenge, fortunately, some subtle structural differences amongst biothiols offer the possibility of achieving specific fluorescent sensing of biothiols. Several sophisticated complex reactions have been proposed to design fluorescent probes for real-time bioimaging of particular thiols in living cells. In this section, we will provide a brief introduction to common strategies for designing specialized fluorescent probes for intracellular biothiols.

### 3.1. Specific Fluorescent Probes for Intracellular Cys

Acrylates are widely used as reactive sites for the specific detection of cysteines. As shown in Figure 6, an electron-deficient acrylate can quench the fluorescence of a free probe, and in the presence of GSH, a thioether can be obtained by Michael addition but without any emission. On the contrary, Cys and Hcy could induce further intramolecular nucleophilic substitution reactions between the primary amidogen and the carbonyl, leading to the release of light fluorophores. Furthermore, the formation of a seven-membered ring structure is kinetically more favored by far, allowing for specific detection of Cys. A large number of acrylate-based probes have been developed so far to monitor intracellular Cys based on various fluorogens in visible and near-infrared regions or with AIE and two-photon characteristics (Figure 7).

The Chen group reported an acrylate-based fluorescence probe **7** (**DPAS-Cys**) to detect Cys in lipid droplets using an AIEgen DPAS [47]. Probe **7** fluoresced at 560 nm when stimulated at 360 nm in the presence of Cys, with a low detection limit of 2.4 μM. Due to its high sensitivity and selectivity for Cys, as well as its superior biocompatibility, probe **7** was effectively employed for fluorescence imaging of endogenous and exogenous Cys in living cells. Further colocalization experiments proved that probe **7** was capable of tracking the Cys in lipid droplets, which was attributable to its high hydrophobicity. The Liu group designed and synthesized a novel fluorescent probe **8** by incorporating two acrylate groups into the flavonoid moiety [48]. Most interestingly, probe **8** exhibited distinct responses to different concentration ranges of Cys. The emission at 509 nm increased within a low concentration range of Cys (0~10 mM, 0~1 equiv) and reached a plateau at a high concentration (>10 mM), while another increased fluorescence at 449 nm was observed in the high Cys concentration range (0~40 mM, 0~4 equiv). The intensity ratio (I_449_/I_509_) increased with the enhancement of Cys concentration with good linearity, indicating a potential to detect Cys by ratiometry. The probe with low cytotoxicity was finally used to differentially visualize Cys at different concentrations in living cells. The Liu group synthesized an acrylate-based ratiometric fluorescent probe **9** (**TIFC**) by utilizing two identical isophorone-malononitrile structures as the fluorophore [49], whose ratio of fluorescence intensity (I_568_/I_540_) was linearly related to the Cys concentrations in the range 0–300 μM, covering the range of the physiological Cys level, with a detection limit of 105.6 nM. With desirable stability and biocompatibility under physiological conditions, probe **9** was successfully applied to measure the endogenous and exogenous Cys in osteoblasts by ratiometric fluorescence imaging. However, the relatively short excitation and emission bands of the aforementioned probes are always implicated in poor penetration depth and large background noise, which are not favorable for tissue and in vivo bioimaging.

Accordingly, an acrylate-decorated anthocyanidin derivative served as a red-emitting fluorescent probe **10** (**HAS**) to selectively detect Cys [50]. Probe **10** responded to Cys in a turn-on manner with an emission at 615 nm when excited at 580 nm, and the fluorescence intensity displayed a remarkable linear connection with Cys concentration. Due to its high membrane permeability and low cytotoxicity, probe **10** was successfully used for fluorescence imaging of Cys in live cells. An electron-withdrawing malononitrile group might chromatically shift excitation/emission wavelengths of dye into the red-emitting region by extending the conjugation and boosting the ICT effect of the D-A structure. Inspired by this, the Jiang group designed a deep red-emitting fluorescent probe **11** (**DRP-Cys**) based on a malononitrile-modified xanthene derivative with acrylate as the Cys recognition site [51]. After being excited at 580 nm, probe **11** displayed a turn-on response to Cys with a 30-fold increase in the fluorescence at 645 nm. Probe **11** obtained a fast response time of about 1 min and a low limit of detection of 10 nM in the Cys range of 0–200 μM, suggesting favorable sensitivity for Cys. Because of its high selectivity, biocompatibility, and membrane permeability, probe **11** was successfully used for fluorescent imaging of Cys in living cells and mouse models. Another strong electron-withdrawing group, 2-dicyanomethylene-3-cyano-4,5,5-trimethyl-2,5-dihydrofuran (TCF), had been reported to produce red-emitting D-A-type fluorophores. Therefore, the Zhu group developed a TCF-based long-range measuring probe **12** (**TCFQ-Cys**) for Cys detection, with acrylate serving as the response group [52]. The free probe **12** exhibited a turn-on response to Cys with a red emission at 633 nm; more excitingly, the emission intensity was linear to Cys concentrations throughout the wide range of 0–300 μM, covering the range of the physiological Cys level (30–200 μM), and the limit of detection was determined at 0.133 μM. Due to its high sensitivity and selectivity for Cys and favorable compatibility, probe **12** was successfully engaged in fluorescence imaging for endogenous and exogenous Cys in cells and Caenorhabditis elegans.

Because of its low background fluorescence, minimal tissue absorption, and deep tissue penetration, near-infrared (NIR) fluorescence with a wavelength in the range of 650–1000 nm is widely used in imaging in vivo. The Qiu group created a novel acrylate-based NIR fluorescent probe **13** (**Cys-TCF**) for selective in vivo bioimaging of Cys using a TCF-tailed NIR fluorophore [53]. Excited at 570 nm, the fluorescence intensity (607 nm) of probe **13** increased along with the enhancement of Cys concentration (0–200 μM) in favorable linearity, and the limit of detection was estimated at 0.04 μM. Owing to its NIR-emitting characteristics, probe **13** proved successful in fluorescence imaging of Cys in live cells and zebrafish. By utilizing a (2-(2′-Hydroxyphenyl) benzothiazo (HBT) derivative integrated with a strong electron-withdrawing isophorone, the Yin group developed another acrylate-based NIR fluorescent probe **14** (**SYP**) for the real-time imaging of Cys in vivo [54]. Attributed to the ESIPT characteristic of HBT, probe **14** exhibited an excellent large Stokes shift of 263 nm with enhanced fluorescence at 686 nm in the presence of Cys, which was beneficial to avoiding background interference. Further optical experiments in buffer revealed that probe **14** was stable and appropriate for the selective detection of Cys at pH ranging from 5.0 to 9.0, with detection sensitivity at pH 7.4 being more sensitive and rapid than at pH 8.0. Due to its favorable sensitivity and biocompatibility, probe **14** was applied to monitor Cys in living cells and tumor mice by fluorescence imaging. Similarly, the Li group created a novel NIR fluorophore by hybridizing xanthene and isophorone and then caged it with an acrylate group to obtain a NIR fluorescent probe **15** (**IX**) for Cys [55]. In the presence of Cys, probe **15** displayed a remarkable increase in fluorescence at 770 nm with a large Stokes shift (180 nm), which is advantageous for biosensing and bioimaging. Probe **15** was successfully applied to the fluorescent imaging of Cys in living cells and HCT116-xenograft tumor mice. Zeng et al. developed a novel near-infrared fluorescent probe **16** (**BHcy-A**) to visualize Cys via heteroatom substitution on the semi-heptamethine dye [56]. Probe **16** may preferentially respond to Cys with near-infrared (NIR) emission (742 nm), which is desirable for bioimaging due to its excellent penetration and high signal-to-noise ratio (S/N). The bioimaging results indicated that probe **16** could specifically accumulate in the lysosomes, revealing its potential to study the function of lysosomes and lysosomal Cys in living cells.

Two-photon fluorescent probes are also a research hotspot in bioimaging, benefiting from strong penetration, minimal cytotoxicity, reduced photobleaching, and high resolution. The Lin group reported a two-photon ratiometric fluorescent probe **17** (**Nap-Cys**) employing naphthalimide as the reporter [57]. Probe **17** specifically responded to Cys over Hcy and GSH; the emission intensity ratio between 550 nm and 440 nm (I_550_/I_440_) is in good linearity with the concentration of Cys, with a moderate limit of detection at 1.80 μM in the range of 10–80 μM. The *p-*toluenesulfonamide serving as an endoplasmic reticulum (ER)-specific group allows probe **17** to detect endogenous and exogenous Cys levels in the ER by two-photon imaging for the first time. Given the advantages of long wavelengths, two-photon fluorescent probes emitting near-infrared light will be more suitable for biological in vivo imaging. The Zhang group developed a red-emitting probe **18** (**ACP**) to achieve the detection of Cys by two-photon fluorescence imaging [58]. Probe **18** demonstrated a fast response, significant intensity enhancement, and high selectivity for Cys over Hcy and GSH. Due to its long-wavelength emission (610 nm), two-photon fluorescence imaging of Cys was obtained in deep tissue up to 120 μm depth, and an increasing Cys was observed in the inflamed mouse model. The Zhou group reported a NIR-emitting two-photon fluorescent probe **19** (**SDP-A**) with an acrylate fraction as the ESIPT/ICT blocker and Cys response site [59]. In the presence of Cys, there was an increase in fluorescence intensity at 713 nm, with a noticeable Stokes shift of 302 nm due to the combined ESIPT and ICT. Probe **19** demonstrated desirable sensitivity and selectivity for Cys over Hcy and GSH, with a limit of detection of 102 nM. The fluorescent scaffold, most crucially, displayed a favorable two-photon absorption cross section (213.5 GM at 820 nm) and high quantum yields (1.52–18.17%), making it appropriate for two-photon fluorescence imaging of biological samples. By two-photon fluorescence imaging with probe **19**, three-dimensional perspective images of the mouse brain were achieved, and finally, the abdominal cavity down to a depth of more than 200 μm was also observed.

To further improve the specificity for Cys, the Li group introduced furan and thiophene into the acrylate sensing group to build new Cys-responsive probes **20** and **21** (Figure 8) [60]. The probes selectively respond to Cys by increasing their fluorescence at 523 nm. Both probes had exceptional fluorescence stability at physiological pH levels, and MTT assays confirmed their low cytotoxicity. Probes **20** and **21** were then applied for fluorescence imaging of endogenous and exogenous Cys in living cells. Finally, the Cys in aged rats chronically treated with APAP (*N*-acetyl-*p*-aminophenol, which consumes Cys by APAP metabolism) was visualized by fluorescence imaging using probe **21**, and the results confirmed the permanent loss of Cys caused by the metabolism of APAP in the liver. It was also proven that Cys loss might aggravate sarcopenia and that the proper supplementation of Cys might be beneficial to the elderly under APAP treatment.

Mustafa Emrullahoğlu’s group utilized propargylate as the reaction site to develop a novel fluorescent probe **22** (**FL-PRP**) [61]. The Cys could react with the propargylate by a cascade reaction involving nucleophilic addition and substitution to uncage the fluorescein dye, affording an increased emission band at 515 nm under excitation at 460 nm (Figure 9). According to the titration experiment, the detection limit of probe **22** was calculated to be 182 nM. With favorable biocompatibility, probe **22** was successfully used for fluorescence imaging of endogenous and exogenous Cys in living cells.

Cys/Hcy but not GSH can undergo a selective addition-cyclization reaction with aldehyde groups, leading to the formation of thiazolidines. Based on this reaction, many aldehyde-based specific fluorescent probes for Cys/Hcy have been developed. The Li group reported an unexpectedly specific dual-channel fluorescent probe **23** for Cys (Figure 10) [62]. Within 15 min of adding Cys, an increased intensity was recorded at 440 nm, and the limit of detection was estimated at 63 nM. The detection of Cys was also shown to be stable and reliable due to its lack of interference from any other amino acids, including Hcy and GSH. However, it is worth noting that DMF, an organic solvent, was used in the detection system with a proportion as high as 50% in the solution, which is unsuitable for bioimaging. Finally, probe **23** was successfully applied to determine the Cys in human serum.

Recently, the Yuan group developed a fluorescent probe **24** (**SPI**) based on a phenothiazine–cyanine scaffold with a sulfamide moiety as a Cys-responsive group [63]. As depicted in Figure 11, probe **24** can respond to Cys via thiolytic reaction to fluoresce at 538 nm with an excitation of 426 nm. Interestingly, the sulfur atom on the phenothiazine can be efficiently oxidized by ·OH into sulfoxide, forming a larger π-conjugation to exhibit red fluorescence (E_x_/E_m_ = 485/608 nm). This dual-site probe **24** demonstrated desirable sensitivity and selectivity towards Cys/·OH over the other species, enabling quantitative detection and real-time imaging of ·OH and Cys in their own fluorescence channels. Finally, the Cys/·OH-regulated redox balance in mice and zebrafish was also monitored by fluorescent imaging with probe **24**, indicating its potential application in the study of physiological and metabolic processes.

### 3.2. Specific Fluorescent Probes for GSH

Sulfone and sulfoxide, which can be nucleophilically substituted by biothiols via the SNAr reaction mechanism, were commonly employed as reactive sites in the construction of fluorescence probes for GSH with high selectivity. For example, the Fang group recently developed a sulfuryl-based fluorescent probe **25** (**R13**) with a naphthalimide skeleton to quantify GSH concentration in living cells and tissues (Figure 4A) [64]. Probe **25** exhibited negligible fluorescence because of the inhibited ICT effect. After adding GSH, a SNAr reaction led to the formation of **Nap-SG**, which exhibited increased fluorescence in the green channel (498 nm) (Figure 4A,B). With excellent sensitivity and moderate selectivity toward GSH in physiological conditions, probe **25** was used to determine GSH levels in mouse livers after X-ray irradiation, revealing that irradiation-induced oxidative stress leads to the depletion of GSH. Encouraged by the results, which were in good agreement with those obtained by HPLC, quantification of GSH content in Parkinson’s mouse brains was achieved by a straightforward fluorometric assay with probe **25**.

The Liu group designed and synthesized a dual-response fluorescence probe **26** (**Mito-NA-BP**) for GSH and its metabolic product SO_2_ based on a FRET scaffold, using chromene as the acceptor and naphthalimide as the donor (Figure 5A) [65]. With 4-fluoro sulfoxide and chromene as the reactive units towards GSH and ${\mathrm{SO}}_{3}^{2-}$, Probe **26** could respond to GSH and ${\mathrm{SO}}_{3}^{2-}$ in distinct patterns via two emission channels at 496 nm (green) and 638 nm (red) by modulating the FRET process (Figure 5B,C). Owing to its favorable performance, including high sensitivity, selectivity, and low cytotoxicity, probe **26** was applied to track exogenous ${\mathrm{SO}}_{3}^{2-}$ and endogenous Cys in real-time, monitor the metabolic process of GSH-SO_2_, and visualize the effect of high-dose SO_2_ on GSH in living cells in real-time.

Boron-dipyrromethene (BODIPY) derivatives with reactive groups, such as DNBS, chlorine, 2,4-dinitrobenzenyl (DNB), and selenium, can also respond to biothiols by SNAr reactions [66,67,68,69]. Chen et al. used a chlorinated BODIPY to create a sequentially activated fluorescent probe **27** (BNS) for visualizing nitric oxide (NO)-induced GSH upregulation in living cells and in vivo (Figure 6A) [70]. Free probe **27** could not react with GSH and showed negligible fluorescence due to a PET process from diamine to BODIPY. In the presence of NO, the diamine group of BNS was transformed into a triazole derivative with enhanced fluorescence at 565 nm (yellow) by the inhibited PET process [29]. Furthermore, the electron-deficient triazole made the three-position in BODIPY electrophilic enough to react with GSH to form a thiol-BODIPY, resulting in a red-shifted fluorescence at 595 nm (red) (Figure 6B). The yellow and red fluorescence intensities showed a good linear relationship with their corresponding analyte concentrations, with favorable selectivity and sensitivity, respectively. Imaging of NO and GSH was originally performed in human umbilical vein endothelial cells (HUVECs), with variations in fluorescence intensities indicating that the production of GSH could be induced by exogenous NO in the living cells (Figure 6C). The further imaging results consistently showed that either pravastatin or Vitamin C (VC) could increase the level of intracellular NO, which in turn induced the activation of *γ*-Glutamylcysteine synthetase (*γ*-GCS) and eventually led to the upregulation of intracellular GSH.

Feng et al. presented a dual-response probe **28** (**MGV**) that can simultaneously detect GSH and viscosity, with 2,4-dinitrobenzenyl (DNB) as the reactive site (Figure 7A) [71]. Probe **28** exhibited absorption at 509 nm and emission at 650 nm (red channel). In the presence of GSH, DNB was removed from the probe, leading to an enhancement of fluorescence at 535 nm (yellow channel), which increased gradually with the smoothly decreased fluorescence at 650 nm (Figure 7A–C). Whereas the increased viscosity could prevent the free rotation of the vinyl group, resulting in an elevated fluorescence at 627 nm when excited at 510 nm. Probe **28** was capable of not just tracking GSH and viscosity in living cells but also visualizing the production of bleb vesicles after nystatin stimulation. Apoptosis induced by cisplatin was finally investigated by fluorescence imaging using probe **28**, with enhancements in both red and yellow channels indicating the upregulation of GSH and viscosity induced by apoptosis.

The Yin group developed a fluorescent probe **29** (**HBT-COU**) to detect GSH in living cells [72], using a hybrid of coumarin and 2-(2-Hydroxyphenyl)benzothiazole featuring ESIPT as the fluorophore. Probe **29** exhibited a fast ratiometric response to biothiols, with a green emission at 519 nm owing to the ICT process of the enol form and another red emission at 621 nm attributed to the ESIPT process of the keto form (Figure 12). With high selectivity towards GSH over Hcy and Cys, probe **29** was then successfully utilized for ratiometric fluorescence imaging of endogenous and exogenous GSH in living cells; declining GSH levels were visualized in apoptotic cell lines compared to the healthy lines.

In addition to the specific fluorescent probes for Cys and GSH, a few selective probes for Hcy have also been reported [73,74], of which we will not give detailed examples here. Readers interested in these can read the corresponding references provided by us.

### 3.3. Mitochondria-Specific Fluorescent Probes for Biothiols

Mitochondria are the main sites of cellular metabolism, and their malfunction frequently causes cellular redox imbalance, which leads to cell death. Biothiols are important reducing agents in mitochondria that may remove oxidative free radicals generated by mitochondria and maintain intracellular redox homeostasis. Therefore, the development of mitochondria-specific probes for biothiols is of practical significance for studying the physiological and pathological functions of biothiols at the subcellular level.

The proton pumps in the inner mitochondrial membrane transport protons into the mitochondrial intermembranous region during mitochondrial respiration, resulting in the creation of a strongly negative mitochondrial transmembrane potential. Inspired by this process, many delocalized lipophilic cations exhibit mitochondrion targeting capabilities. One typical targeting ligand is triphenylphosphonium (TPP). Utilizing it as the mitochondria targeting group, Zhang et al. reported a dual-channel fluorescent probe **30** (**NTG**) for simultaneous detection of GSH and ONOO^-^ in mitochondria [75]. Probe **30** could respond to GSH and ONOO^-^ in two channels; an enhanced fluorescence was recorded in the red channel (670 nm) in the presence of GSH, while another increased emission was found in the green channel (530 nm) in the simultaneous presence of GSH and ONOO^−^, whereas ONOO^−^ could afford only weak fluorescence in the green channel (Figure 8A,B). Probe **30** displayed desirable selectivity, sensitivity, stability, and mitochondria targeting (Figure 8C), which was then applied to track endogenous and exogenous GSH and ONOO^-^ in normal, inflammatory, and tumor cells of different stages. Imaging experiments were also carried out to assess the capability of discriminating different zebrafish models. Significantly enhanced green fluorescence and sharply decreased red fluorescence were found in inflammatory zebrafish compared with the normal ones, while increased green fluorescence and unchanged red fluorescence were recorded in cancerous ones (Figure 8D).

On the basis of a positively charged cyanine fluorophore, Zhu et al. developed a novel mitochondria-targetable probe **31** for biothiols (**Cy-DNBS**), with the DNBS moiety serving as the reactive site (Figure 13) [76]. Probe **31** showed weak fluorescence under excitation at 545 nm; after adding biothiols, a tremendous increase was found at 604 nm with a moderate Stokes shift of 59 nm, and the intensity at 604 nm was used to estimate the level of biothiols. Further experiments demonstrated the probe’s favorable properties, including higher sensitivity and selectivity to biothiols over other biological species, excellent photostability, low cytotoxicity, and mitochondria targeting. Probe **31** was then used for fluorescent imaging of endogenous biothiols in living cells, and attenuation of biothiols was observed in HeLa cells pretreated with hydrogen peroxide, demonstrating its potential to track intracellular biothiol changes under oxidative stress.

Lin reported another cyanine-based NIR fluorescent probe, **32** (**GalCys**), to monitor levels of Cys in mitochondria (Figure 11) [77]. The probe exhibited significantly enhanced fluorescence emission at 668 nm after responding to Cys (0–200 μM). With outstanding sensitivity and selectivity to Cys, desirable NIR emission, and favorable biocompatibility, **GalCys** was applied to detect changes in Cys in mitochondria of living cells, proving for the first time that the development of diabetes led to oxidative stress in mitochondria. A dramatic decrement of Cys was observed in further imaging experiments on mouse models exposed to excess particulate matter from environmental pollution, indicating serious destruction from airborne particulate matter pollution to the redox balance of organisms.

In recent years, multi-targeting fluorescent probe**s** have demonstrated great potential to monitor the interplay between different organelles and the imaging of invisible biophysical parameters. The Song group designed and synthesized a Cys-responsive fluorescent probe **33** (**BEB-A**), which can label mitochondria due to the benzoindolium cationic group serving as a targeting group [78]. In the presence of Cys, probe **33** exhibited an elevated emission at 616 nm under excitation at 550 nm, with a limit of detection of 0.027 μM. Most importantly, after responding to the Cys, probe **33** could enter the nucleus and bind to the RNA, leading to an enhancement in red fluorescence and providing a new approach for the study of mitochondria and nucleolus. Similarly, the He group developed another Cys-responsive mitochondrion-targetable probe **34** (**TSQC**) using acrylate-decorated cationic AIEgen [79]. Free probe **34** emitted at 750 nm, whereas after adding Cys, the probe could spontaneously target lipid drops with an enhanced emission at 650 nm, indicating its ability to monitor LDs and mitochondria in dual channels. Using Probe **34**, apoptosis induced by LPS, H_2_O_2_, or UV light exposure treatments was tracked by dual-channel visualization. Finally, apoptosis associated with acute and chronic epilepsies was also monitored in mouse models, providing excellent applicability for Cys-activatable imaging of LDs and mitochondria in two channels for the first time.

## 4. Design Strategies for Reversible Fluorescent Probes for Biothiols

Biothiols, as essential regulators in living systems, play critical roles in various physiological processes such as redox balance maintenance, signal transduction, gene expression, protein synthesis, and so on. Real-time monitoring of dynamic changes in biothiols is of great significance to reveal their physiological and pathological functions and provide a foundation for early treatment and diagnosis of diseases. To this end, many reversible fluorescent probes have been reported to monitor the dynamics of GSH in living specimens recently, and these reversible fluorescent probes are frequently designed on the basis of the reversible Michael addition reactions (Figure 14) [80,81].

The Bhuniya group reported a reversible ratiometric fluorescent probe **35** (**GS_cp_**) for imaging GSH in living cells (Figure 14 and Figure 15) [82]. The probe features a coumarin-vinyl-pyridine fluorophore that specifically responds to GSH over other biothiols. The absorption shifts from 410 to 350 nm and fluorescence shifts from 510 to 460 nm, and the ratio of fluorescence intensity between 460 nm and 510 nm (I_460_/I_510_), were linearly correlated with the increasing level of GSH. More interestingly, probe **35** demonstrated reversible sensing in a GSH-H_2_O_2_ redox dynamic system with a dissociation constant (Kd) of 2.47 mM under physiological conditions. The high selectivity, sensitivity, and favorable biocompatibility of probe **35** allowed for ratiometric fluorescent images of GSH in living cells, and its capability of nucleoli targeting was discovered.

The Liu group developed a novel reversible probe **36** (**RP-2**) for detecting GSH in living cells (Figure 14) [83]. Free probe **36** showed a fluorescent emission of 587 nm under excitation at 530 nm. On the titration of GSH, a new fluorescence appeared at 505 nm when excitation was at 420 nm, while negligible change was found in the fluorescence at 587 nm. Optical experiments indicated that the intensity ratio of I_505_/I_587_ was competent to detect GSH ranging from 0.1 to 10 mM, while the intensity of I_505_ was more suitable for GSH with a larger concentration range (0.1 to 30/50 nM). Considering the level of intracellular GSH expression, the former could be used to detect intracellular GSH. Most importantly, probe **36** exhibited a reversible and extremely fast response toward GSH (half-time 3 s), which was necessary for real-time imaging. Due to its favorable biocompatibility, probe **36** was successfully used for ratiometric fluorescent imaging in living cells, and an image processing program was developed in MATLAB (a commercial mathematics software) to facilitate the real-time imaging of GSH.

Shen et al. reported a reversible fluorescent probe **37** (**B-GSH**) based on a BODIPY derivative with exocyclic extended conjugations (Figure 14) [84]. The addition of GSH afforded an increased absorption at 527 nm with enhanced fluorescence at 544 nm (green channel) and a decreased absorption at 594 nm with reduced fluorescence at 603 nm (red channel). Additionally, reversible behaviors were verified after the elimination of GSH. Finally, intracellular GSH dynamics were monitored by fluorescent imaging using probe **37**. Pretreated with cisplatin, GSH concentration first increased to the maximum, then fell back to the initial concentration without significant apoptosis in cisplatin-resistant A549 cells, while GSH levels increased until cell death in cisplatin-sensitive cell lines. The results effectively proved the potential of probe **37** to detect intracellular GSH dynamics in response to therapeutics.

The Cui group developed a novel reversible NIR ratiometric fluorescent probe **38** (**EpSiP**) by molecular engineering, enabling in vivo quantification of GSH in intact tissues and animals (Figure 14) [85]. Probe **38** was constructed on a FRET scaffold, where Si-rhodamine and phospha-rhodamine were used as the FRET acceptor and donor, respectively. An effective FRET process was recorded in the NIR region, and the absorption and emission spectra of the two fluorophores were well resolved (Δ*λ*_abs_ ≈ Δ*λ*_em_ = 61 nm), which was conducive for bioimaging. Free probe **38** exhibited a strong emission in the red channel (736 nm) belonging to the phospha-rhodamine and a weak emission in the green channel (675 nm) belonging to the Si-rhodamine under an excitation of 640 nm. In the presence of GSH, the intensity in the red channel decreased with an enhancement in the green channel, and the spectra recovered immediately after adding H_2_O_2_, indicating a reversible response to GSH. Probe **38** exhibited favorable reaction kinetics and thermodynamics to GSH (K_d_ = 4.9 mM, *k* = 81 M^−1^s^−1^, t_1/2_ = 0.57, [GSH] = 10 mM), and was then applied to quantitatively detect the GSH levels in different biospecimens, including living cells, xenografted tumors on the mouse, chronic renal failure, and liver fibrosis in time.

The Yin group designed a reversible mitochondria-targeted fluorescent probe **39** (**Mito-1**, Figure 3) based on a coumarin derivative to detect changes in mitochondrial Cys (Figure 14) [86]. Michael addition shortened the conjugation system of probe **39**, which led to a blue shift in the UV-vis absorption and an enhancement in fluorescence intensity at 498 nm. Due to its advantages, including high sensitivity, fast response, excellent reversibility, and low cytotoxicity, probe **39** was successfully used for the fluorescent detection of mitochondrial Cys in living cells. Encouraged by these results, another lysosome-targeting probe named probe **40** was also designed. The two probes were used to study the activation of the cellular inherent antioxidation system by resveratrol, and the enhancement in fluorescence indicated the Cys upregulation induced by the resveratrol.

## 5. Design Strategies of Fluorescent Probes for Discriminated Detection of Biothiols

In the above sections, we have introduced specific and reversible fluorescent probes. Most of these probes have been applied to monitor biothiols in cells and in vivo, showing potential applications in biomedical and disease research. However, they always fail to provide researchers with detailed information on diverse biothiols, as they cannot simultaneously detect multiple biothiols. Of course, this dilemma can be overcome by using multiple probes, but such multi-probe strategies often result in distorted detection results due to the differences in biothiol distribution, metabolism, and membrane penetration. Therefore, it is of great significance to construct discriminating fluorescent probes that can simultaneously detect and discriminate among multiple biothiols [87,88].

### 5.1. Discriminated Probes with a Single Reactive Site for Biothiols

7-nitrobenzofurazan (NBD) is a well-known reactive group involved in the design of specific and discriminating fluorescent probes for biothiols. Due to its strong electron deficiency, NBD quenches the fluorescence of the probe efficiently by the PET process. Reactions with biothiols result in the formation of free fluorophores with enhanced emission; specifically, reactions between NBD-probe and GSH lead to the formation of thiol-NBD, while amino-NBD derivatives are generated with Cys/Hcy, exhibiting distinct optical performance from thiol-NBD to amino-NBD. Therefore, a systematic approach can be achieved for distinguishing detection towards GSH and Cys/Hcys (Figure 16). In 2021, Wang et al. reported a probe **41** (**MZ-NBD**) for the detection of GSH in living cells, with NBD as a fluorescent response group (Figure 16) [89]. Hcy and Cys reacted with probe **41** by cascade nucleophilic substitution, leading to the formation of amino-NBD with fluorescence at 560 nm, whereas GSH and probe **41** afforded thiol-NBD by a one-step nucleophilic substitution with slightly red-shift fluorescence at 565 nm. Noteworthy, probe **41** could detect GSH with fluorescence fusion at 530 nm, while two distinct emission peaks were found in the detection of Hcy and Cys. Owing to its high sensitivity, favorable selectivity, fast response, and moderate biocompatibility, probe **41** was used to track the endogenous and exogenous GSH in MCF-7 cells, and its ability to avoid interference from Cys was also verified in the imaging experiments. The Feng group developed a unimolecular fluorescent probe **42** (**ZED**) based on a coumarin-rhodamine FRET platform with NBD and hydrazide as the Cys/Hcy and HClO sensing sites, respectively (Figure 16) [90]. With imidazolium used as the targeting group for mitochondria, probe **42** was applied to track the Cys/Hcy, HOCl, Δψm, and opening of the MPT pore by four channels without any crosstalk.

Although chemotherapy is an effective method to treat cancer, it remains controversial whether drug therapy can lead to abnormal expression of reactive oxygen species (ROS) and reactive sulfur species (RSS) by increasing intracellular oxidative stress. The Ye group developed a multifunctional probe **43** (**RSS-HClO**) to track biothiols and HClO levels during chemotherapy (Figure 17) [91]. Probe **43** was constructed with a coumarin-hemicyanine moiety and NBD as the reporter and reactive site, allowing it to distinguish Cys/Hcy, GSH, and SO_2_ using three channels simultaneously. Despite almost no fluorescence due to the PET effect, probe **43** demonstrated a substantial increase in the red channel when exposed to GSH. On the other hand, either the red or green channel recorded significantly enhanced fluorescence after Cys/Hcy addition, while HClO only caused an increase in the fluorescence blue channel (Figure 17). Attributed to its high sensitivity, selectivity, and favorable biocompatibility, probe **43** was successfully utilized to discriminatively visualize Cys/Hcy, GSH, and HClO, and its mitochondria targeting ability was verified by co-localization experiments. Further, an upregulation of cellular biothiols and HClO in tumor cells during chemotherapy was observed, indicating potential applications in studying pharmacologic actions.

The Tang group has designed and synthesized a fluorescent probe **44** (**TAT-probe**, Figure 18) for discriminating imaging of biothiols based on a native chemical ligation (NCL) reaction, using the penetrating peptide TAT (RRQRRKKRG) as a mitochondrial targeting unit [92]. Probe **44** was constructed in a FRET form, with naphthalimide serving as the donor, rhodamine as the acceptor, and thiophenol ester as the sensing site and linker between the donor and acceptor. When excited at 404 nm, probe **44** exhibited a red fluorescence peaking at 585 nm in the absence of biothiols due to the FRET process. However, upon the addition of biothiols, the red fluorescence fell with an enhanced green emission at 520 nm, indicating the blocked FRET process, and the fluorescence intensities ratio (I_520_/I_585_) showed good linearity to the concentration of biothiols, making this ratiometric fluorescence probe less prone to interference from the concentration and distribution of the probe. More excitingly, probe **44** exhibited almost unchanged red fluorescence towards GSH under excitation at 545 nm, while a significant decrease in red fluorescence was observed in Cys/Hcy titration, enabling probe **44** to sensitively distinguish GSH from these three biothiols. The probe was further utilized for fluorescent imaging of biothiols in living cells, with favorable compatibility and mitochondrial targeting from the TAT peptide. Most importantly, GSH discrimination was achieved in NEM-pretreated cells.

The Song group reported a multi-response site probe **45** (**KC**) to discriminatively visualize thiols in living cells and zebrafish (Figure 19) [93]. The probe utilized an azide moiety as a reactive site for H_2_S, causing a pronounced increase in the red channel. The thiocoumarin was applied to distinguish thiols based on different reaction mechanisms. GSH affords a sharp enhancement in the blue channel via the SNAr reaction, while Cys undergoes a significant promotion in the green channel by a cascade reaction that includes SNAr reaction and Smiles rearrangement. Hcy follows a similar process to Cys, but its slower rearrangement results in a dual increase in blue and green channels. Probe **45** displays moderate sensitivity, favorable selectivity, and low cytotoxicity, making it suitable for detecting thiols in living cells and three day old zebrafish models.

The Wang group has developed a novel fluorescence probe **46** (**CySI**) based on cyanines and thioesters capable of distinguishing Cys and Hcy through two distinct emission channels under single excitation (Figure 20) [94]. When excited at 550 nm, the probe displayed enhanced fluorescence at 625 nm and 740 nm in reaction with Cys and increased fluorescence at 740 nm for Hcy, without any fluorescence response to GSH. This feature enables the simultaneous identification and determination of Cys and Hcy through two separate channels. Further, the probe was used to visualize exogenous Cys and Hcy in living cells, demonstrating its capability to effectively target mitochondria and its potential to monitor endogenous Cys and Hcy fluctuations in mitochondria through red emission channels.

### 5.2. Discriminated with Multiple Reactive Sites

The Song group developed a ratiometric probe **47** (CP) based on the conjugate addition-cyclization reaction (Figure 21) [95]. When the biothiols were added, probe **47** exhibited two distinct fluorescence bands upon a single excitation at 380 nm. Interestingly, Cys caused a single fluorescence in the blue channel (485 nm), whereas dual emissions in the blue (608 nm) and red channels (485 nm) were recorded in the detection of Hcy. Due to its high sensitivity, high selectivity, low cytotoxicity, and outstanding photophysical properties, probe **47** was successfully applied to ratiometric fluorescence imaging of Cys and Hcy in HeLa cells and zebrafish, with effective discrimination between Cys and Hcy.

The Yin group designed and synthesized a fluorescent probe **48** (**CBB**) for visualization of endogenous biothiols using the 4-chlorine moiety (Figure 22) [96]. Probe **48** responded to biothiols with different mechanisms, affording simultaneous discrimination between Cys, Hcy, and GSH via three distinct emission channels. Owing to its favorable sensitivity, selectivity, and biocompatibility, probe **48** was employed for discriminative imaging of endogenous GSH, Hcy, and Cys and their transformation in living cells.

The Liu group reported a fluorescent probe **49** (**YF**) for distinguishing the visualization of endogenous biothiols using a coumarin-hemicyanine fluorophore (Figure 23) [97]. Probe **49** was equipped with three reactive sites and has multiple optical channels for fluorescence detection. Different sensing processes were recorded in the optical spectra when adding biothiols. Probe **49** exhibited similar trends in fluorescence to the three biothiols, but less amplitude was observed in GSH addition. In the time-dependent emission spectra, two-phase change processes were recorded for GSH, while three phases were found for Cys/Hcy, which was consistent with their varied response mechanisms. Therefore, it is possible for probe **49** to simultaneously discriminate GSH from Cys/Hcy. Due to its favorable selectivity, permeability, and low cytotoxicity, probe **49** was successfully employed to discriminate Cys/Hcy and GSH in A375 cells simultaneously based on different signaling patterns, and biothiols in fetal bovine serum samples were also measured by probe **49** with results consistent with those obtained by the Ellman method.

As important biological mercaptans, Cys, Hcy, and GSH transform each other and participate in many critical physiological processes such as signal transduction, metabolism, and the maintenance of redox homeostasis. Studies have indicated that abnormal levels of sulfur dioxide (SO_2_) are closely related to the occurrence and development of cancer. Since intracellular SO_2_ is primarily generated through two distinct pathways, GSH and Cys, it is desirable to develop distinguishable fluorescent probes to trace these metabolic pathways and improve disease treatment. To concurrently follow these metabolic pathways, the Yin group developed probe **50** with multiple responding sites, including a constructed site in the reaction process (Figure 24) [98]. Probe **50** could respond sequentially to both biothiols and SO_2_ and afford separate fluorescence signals to Cys/Hcy and GSH, allowing them to differentiate the two metabolic processes. In the Cys/Hcy pathway, two enhanced fluorescence bands were found in the blue and red channels in the presence of Cys/Hcy, and the red channel fell with an increase in the blue channel after adding SO_2_, while in the GSH pathway, two elevated fluorescence bands were observed in the green and red channels in the presence of GSH, and both channels receded after adding SO_2_. Probe **50** was successfully applied to fluorescence imaging of Cys/GSH metabolism in living cells and HeLa tumor nude mice. Highly expressed thiols were observed in tumors compared to healthy tissues, and more excitingly, the probes performed efficient tumor targeting due to a benzopyrylium intermediate in the GSH pathway.

The Chen and Song groups described a novel fluorescent probe **51** (**RC**) using the combination of resorufin and coumarin moieties (Figure 25) [99]. In the presence of thiols, a S_N_Ar substitution process cleaved off the resorufinyl group from the coumarin with a red emission, leading to the formation of thiol-coumarins. Subsequently, these thiol-coumarins underwent possible Smiles rearrangement and cyclization reactions, yielding diverse products and unique fluorescence combinations via three emission channels. Showing excellent selectivity, sensitivity, and low cytotoxicity, probe **51** was then applied to confocal imaging in living cells, indicating its robust potential to distinguish different cellular thiols.

## 6. Conclusions and Outlook

In conclusion, we reviewed the recent progress of biothiol fluorescent probes over the past four years and briefly introduced the strategies for biothiol probe design from four perspectives: response mechanism, specific imaging, reversible detection, and discriminated detection (Table 1). Over the past decades, fluorescent probes have been intensively studied for the detection of biothiols. Numerous fluorophores featuring red emission, a large Stokes shift, AIE characteristics, and high fluorescent quantum yields have been ingeniously developed and employed in the design of biothiol fluorescent probes [25,26,100]. Diverse imaging modalities, such as turn-on, two-photon, and ratiometric imaging, have also been applied to the biosensing of biothiols. Fluorescent imaging for biothiols has been successfully achieved on a spectrum of specimens, including food, serum, living cells, tissues, organs, and even animal models, which indicates the great potential of probes in pathological research, biomedicine, and disease diagnosis.

However, there are still challenges to studying novel biothiol probes. (1) Although many specific biothiol fluorescent probes have been developed, their selectivity is still insufficient for bioimaging considering the differences in concentration of diverse biothiols in the body. Moreover, compared with specific fluorescent probes for GSH and Cys, few specific fluorescent probes could be used for HCy detection. Therefore, it is vital to explore response mechanisms with efficient selectivity for biothiols. (2) Reversible and discriminant fluorescent probes have been applied to the detection of biothiols in living cells and animal models. However, due to the influence of poor stability, photobleaching, and fast metabolism, it is still difficult to carry out long-term dynamic detection for biothiols in vivo. These drawbacks may be addressed by developing fluorescent dyes or nanoprobes with improved performance. (3) Fluorescence probes with NIR- I/NIR-II emission and two-photon imaging have been applied to detect biothiols in deep tissues in vivo, but the imaging depth is still limited. Therefore, there is still an urgent need to develop multi-modality fluorescent probes combined with NMR, PET, and photoacoustic imaging to provide holistic information on biothiols in vivo for physiological and pathological studies [101,102]. (4) Due to the ubiquitous presence of biothiols in living systems, it is necessary to develop fluorescent probes with controllable responsiveness to achieve high spatiotemporal resolution imaging of biothiols in vivo, such as photocontrollable fluorogenic probes [103,104]. (5) Fluorescence-guided surgery is a robust method to selectively remove tumor sites with the assistance of fluorescence imaging. However, the existing fluorescent probes may have serious adverse effects on the human body. Therefore, developing non-toxic biothiol fluorescent probes would be beneficial for fluorescence-guided surgery [105].

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
