# Peer review of "Recent Progress in the Rational Design of Biothiol-Responsive Fluorescent Probes"

_molecules, 2023, doi:10.3390/molecules28104252_

Round 1

Reviewer 1 Report (Previous Reviewer 1)

The questions raised have been well addressed and I recommend publication of the manuscript.

The Quality of English Language has improved.

Reviewer 2 Report (Previous Reviewer 2)

Altogether the revision of the manuscript increased its quality. With regard to the scientific content the authors present an acceptable revised manuscript. In the rebuttal, the authors discuss the critical points raised by the referees, and they explain the resulting changes in the manuscript. In some cases I still disagree with the way changes were made or rather not made; but this is only a matter of taste, and not of scientific quality.

Specifically, I disagree that this review serves its purpose to be "meaningful for junior researchers to comprehend the sensing and design principles of these probes" because the didactic approach (outline, order principle, choice of only representative examples with clear explanations of the relevant photophysical background) is still not convincing (see comments on the first version) and the pure compilation of literature examples may be more confusing than helpful. But maybe that is too old-fashioned an expectation for a review article.

I therefore still see room for improvement (from my very subjective point of view). But I do not have the time to proofread the review on a weekly basis, so I shall be fine with it and recommend publication.

This manuscript is a resubmission of an earlier submission. The following is a list of the peer review reports and author responses from that submission.

Round 1

Reviewer 1 Report

In this review, the authors summarized the recent progress in the rational design of biothiols-responsive fluorescent probes and focused on the rational design of biothiol fluorescent probes including reaction mechanism, discriminated detection, reversible response, and specific imaging. Overall, the authors provided new ideas for the development of biothiols-responsive fluorescent probes. However, the author's writing is poor, with a large number of grammatical errors, spelling mistakes, missing units, and incorrect/irregular writing. In addition, the pictures in the text need to be improved, and the writing logic needs to be further improved. I think the paper needs to be carefully revised. Some questions and suggestions are listed as follows:

1. There are many grammatical mistakes, especially in some long sentences. It is recommended that the authors find a professional agency to polish the language of the manuscript.

2. Abbreviations that appear for the first time should have full names, such as “Cys”, “GSH”, and “Hcy”.

3. Some proprietary words, such as “no-fluorescence”, are used inaccurately.

4. Some words need to be italicized, such as “in vivo”.

5. The authors should more thoroughly go through the whole text trying to eliminate font format errors, such as "... in the range 0-300 μM, covering the range of the physiological Cys level (30-200 mM)..."; "... range of 0-300 mM, covering the range of the physiological Cys level (30-200 mM), and the limit of detection was estimated at 0.133mM..."; "... The intensity was recorded at 440 within 15 minutes..."; "... probe 31 was applied to track exogenous SO3 and endogenous Cys..."; "... on biothiols in vivo, e.g. photocontrollable fluorogenic probes...". Confusing words like "MATLAB" and "TAT".

6. The format of references is not uniform, and it does not meet the requirements of "molecules". The authors should pay attention to the format of the references.

7. The attached pictures are not clear and need to be replaced with clearer ones, such as Fig. 3B and 3C, Fig. 4, Fig. 5A and 5B, Fig. 6B, Fig. 7B and Fig. 8.

8. The reversible mechanism of Scheme 15, NEM seems incorrect, please review the original text carefully.

9. The authors should make a table in order to show the advantages of reported probes more easily.

Reviewer 2 Report

Xie et al present an extensive compilation of fluorescent probes that may be employed for the detection of sulfide-containing analytes (strangely called "biothiols", that is, a term that I could find in my biochemistry textbooks)). Although this is certainly a relevant and topical subject and the chosen/presented examples are significant contributions to the field, I am not convinced that this "review" would add significantly to the reception of the topic in the scientific community. In general, I do not understand the principle with which the probes were ordered. Without a strict and more obvious, less generic organizing principle the compilation of examples looks rather loose and arbitrary. As a more important point, it is questionable whether this "review" provides particular added value, as compared with the very many existing reviews on this topic.

In addition, some representative major shortcomings of the manuscript are listed exemplarily below:

1.

In the Introduction it is stated: "However, the traditional methods mentioned above ... are not competent to real-time and noninvasive detection to biothiols."

This is no good argument for fluorescent probes as they are also invasive.

2. Quality of graphical material

– Figure 1: this cartoon is meaningless because from these structures alone the relationship to the claimed "design principle" cannot be deduced.

– Figure 2 is meaningless. In addition, it should be noted that it is not the mechanism that is applied, but the reaction itself.

– The meaning of the green arrows in some schemes is not obvious.

– In several captions it is claimed that a "response mechanism" is shown, but there is no mechanism.

3. References:

– An important current review on the topic is not cited: Chem. Sci., 2021, 12, 1220.

– Only two references (19,20) are not sufficient to support the statement "Fluorescent probes have attracted considerable attention ..."

– Too many formal mistakes in the references list.

– Unless this research field is only investigated in one particular region of the world I tend to consider the list of references a bit "unbalanced".

4. Chapter 2

The term "mechanism" is very often mentioned but only in few cases it is explicitly shown or described. And even then, the mechanistic discussion is not appropriate.

5. Recognition

In many cases it is claimed that the probe acts by recognition of the analyte, which is not really true because only few examples really relate to host-guest / ligand-receptor chemistry. In fact, most examples just refer to a chemical reaction between suitable functionalities which is far beyond receptor activity.

6. Fluorophores

– The nature and choice of fluorophores has to be explained.

– The different photophysical properties of the shown fluorophores and the resulting advantages/disadvantages for application should be discussed more systematically, ideally summarized in a meaningful table.

– To be comparable in a series of compounds with different emission ranges Stokes shifts must be given in energy scale.

7.

Please explain the color codes used throughout the manuscript, if there are any.

8.

In many presented application of cell analysis, the pre-treatment of the cells with various different methods is mentioned, but unfortunately not well explained.

9.

Language/Style

– The text is not well written and contains too many typos; plus technical terms are not always correctly used (e.g., rotary olefinic bond, solo probe, fluorescence resonance energy transfer (FRET) scaffold, to overcome some awkwardness in a measurement, specific versus selective, accepter, solo SNAr reaction, Smile rearrangement).

– Definition of abbreviations is not practices consistently.

10.

My plagiarism software indicated that some sentences have been taken verbatim from the original papers (however, overall to an acceptable extent).

Round 2

Reviewer 1 Report

This manuscript is greatly improved after the revisions made by the authors. However, some of the previous comments were not addressed. The following revisions need to be incorporated before publication: please change some long sentences into short sentences, because grammatical errors are still existï¼› the attached pictures are not clear and need to be replaced with clearer ones, such as Fig. 4, Fig. 5B, Fig. 7B and Fig. 8B.

Reviewer 2 Report

I am very sorry, but this is not an appropriate revision.

Only very few substantial changes and corrections were made relative to the many critical points raised in the reviewers' reports, so that most of my critical points, as raised in my first evaluation, are still valid. Indeed, some of the reviewers' comments were addressed appropriately. But unfortunately the authors decided to prepare a very quick and shallow revision instead of a reasonable substantial correction and reorganization.